# Spatial analysis of unimproved drinking water source in East Africa: Using Demographic and Health Survey (DHS) data from 2012–2023

**Lidetu Demoze**[1]*, **Kassaw Chekole Adane**[2], **Jember Azanaw**[1], **Eyob Akalewold**[3], **Tenagne Enawugaw**[4], **Mitkie Tigabie**[5], **Amensisa Hailu Tesfaye**[1,6], **Gelila Yitageasu**[1]

**1** Department of Environmental and Occupational Health and Safety, Institute of Public Health, College of Medicine and Health Sciences, University of Gondar, Gondar, Ethiopia, **2** Department of Environmental and Occupational Health and Safety, College of Medicine and Health Sciences, Wollo University, Dessie, Ethiopia, **3** Department of Epidemiology and Biostatistics, Institute of Public Health, College of Medicine and Health Sciences, University of Gondar, Gondar, Ethiopia, **4** Department of Human physiology, School of Medicine, College of Medicine and Health Sciences, University of Gondar, Gondar, Ethiopia, **5** Department of Medical Microbiology, School of Biomedical and laboratory science, College of Medicine and Health Sciences, University of Gondar, Gondar, Ethiopia, **6** Institute for Sustainable Futures, University of Technology Sydney, New South Wales, Sydney, Australia

* lidetudemoze12@gmail.com

## Abstract

### Background

According to the WHO/UNICEF Joint Monitoring Programme, unimproved drinking water sources include unprotected wells and springs, surface water (e.g., rivers, lakes), vendor-provided water, bottled water (without improved alternatives), and tanker truck-provided water. In East Africa, 68.7% of water at very high risk comes from such sources. Using unimproved drinking water sources can lead to serious health risks, including waterborne diseases such as cholera, dysentery, typhoid fever, and hepatitis. Therefore, this study aimed to map the spatial distribution of unimproved drinking water sources in the region.

### Methods

We analyzed recent Demographic and Health Survey (DHS) data from 12 East African countries (Burundi, Comoros, Ethiopia, Kenya, Madagascar, Mozambique, Malawi, Rwanda, Tanzania, Uganda, Zambia, and Zimbabwe). A total of 206,748 households were sampled in 12 East African countries. Data management and analysis were carried out in several stages, beginning with data cleaning, followed by statistical weighting and data merging. This was then followed by geospatial analysis and mapping, and finally, spatial cluster detection.

### Results

Spatial clusters of unimproved drinking water sources were identified within the study area (Global Moran's I: 0.018, z-score: 87.10, $p < 0.05$). A total of 167 significant spatial windows containing primary and secondary clusters were identified. The first spatial window

**Data availability statement:** The data that support the findings of this study have been deposited in the DHS Archive available from https://dhsprogram.com/data/available-data-sets.cfm.

**Funding:** The author(s) received no specific funding for this work.

**Competing interests:** The authors have declared that no competing interests exist.

contained the primary clusters, while the remaining 166 spatial windows contained secondary clusters. Primary clusters were found in Madagascar and coastal Mozambique, with secondary clusters distributed across all 12 countries analyzed.

## Conclusions

This study identified significant clusters, hotspots, and outliers (high-high clusters) of unimproved drinking water sources across various East African countries. To address these issues effectively, priority should be given to the identified clusters, hotspots, and high-high clusters. Primary recommendations include expanding water treatment facilities, improving water distribution systems, and protecting drinking water sources. Secondary recommendations emphasize strengthening regulations, conducting research, and fostering public-private partnerships to ensure sustainable access to clean water. Finally, we urge collaboration among governments, international organizations, and NGOs to enhance water infrastructure. Their efforts should focus on providing technical assistance, financial support, capacity building, project implementation, advocacy, and financing for drinking water infrastructure in the region. Further research integrating health outcome data with spatial analysis could help identify high-risk regions where the impacts of unimproved water sources are most pronounced.

## Introduction

Access to drinking water is a fundamental necessity and a human right essential for the dignity and health of everyone[1]. Clean, safe drinking water is a basic human right and should be readily available year-round, in sufficient quantities, uncontaminated by chemicals or harmful microbes, and safe for consumption [2]. Improved water sources are defined by WHO/UNICEF in 2000 as those protected from external contamination, especially fecal matter [3,4]. These sources include household connections; public standpipes; boreholes; protected dug wells; protected springs; and rainwater collection [3,4]. In contrast, unimproved water sources include unprotected wells, unprotected springs, surface water (e.g., rivers, dams, or lakes), vendor-provided water, bottled water (unless no other water source is available), and tanker truck-provided water [3–6].

WHO and UNICEF report in 2017 reveals that approximately 3 in 10 people worldwide, or 2.1 billion, lack access to safe, readily available water at home[7].In addition, around 1 in 4 people lacked safely managed drinking water in their homes in 2020[8]. Half of the world's population that drinks water from contaminated sources is found in Africa, despite the MDGs' successes in various regions, Africa still accounts for half of the global population using contaminated water sources[9].According to the WHO reports Sub-Saharan Africa is experiencing the slowest rate of progress in the world[10]. Only 54 percent of people had access to improved drinking water, with just 25 percent in areas experiencing ongoing conflict and political instability[8]. A study conducted in 2020 using secondary data estimated that 29% of the 357,979 households in Sub-Saharan Africa used unimproved water for drinking purposes [11]. In addition, crosssectional survey estimated that in 38.0% of the population of Sub-Saharan Africa SSA (excluding Eritrea and Botswana) reported using an unimproved drinking-water source, although at national levels this was seen to vary between as high as 88.5% in rural Somalia populations to 1.0% in urban populations in Namibia[12].

While water access has improved globally, sub-Saharan Africa still faces significant challenges, with East Africa reporting 68.7% of its water from unimproved sources [13]. Surveys indicate

variability in the use of unimproved water sources, with 47% of the population in Ethiopia, 37% in Kenya, 39% in Tanzania, and 41% in Uganda relying on such sources[14]. Climate change, increasing water scarcity, population growth, demographic changes and urbanization pose significant challenges for improved water source supply systems in East Africa[15]. Moreover, the time spent collecting improved water, often by women and children, limits opportunities for education and economic activity, reinforcing poverty and gender inequality[16]. The use of unimproved drinking water sources can have significant adverse health effects. These include waterborne diseases such as cholera, dysentery, typhoid fever, and hepatitis. Additionally, they can lead to intestinal infections, which can be severe and potentially fatal, particularly in children[17,18]. These groups are more susceptible due to limited access to clean water, poor sanitation, and weakened immune systems, highlighting the urgent need for improved water access and infrastructure[19]. Unimproved water sources can also lead to parasitic infections such as Giardia and Cryptosporidium, causing gastrointestinal issues[20]. Finally, using unimproved water source can result in skin infections and other dermatological conditions[17,18]. Previous studies focused on pooled prevalence and associated factors but lacked insights into spatial variation, which this study aims to address [21]. Thus, conducting a spatial analysis of unimproved water sources will identify high-risk areas and help understand spatial distribution patterns. This analysis can prioritize regions for interventions by correlating the locations of unimproved water sources with incidences of waterborne infections. This study aims to map the spatial distribution of unimproved drinking water sources across 12 East African countries using DHS data from 2012-2023.

## Materials and methods

### Study area, design and data source

The study was conducted in East Africa. East Africa is a vast and diverse region encompassing the Horn of Africa and the eastern fringes of the Sahara Desert[21]. There are a total of 19 East African countries that includes Burundi, Comoros, Djibouti, Eritrea, Ethiopia, Kenya, Madagascar, Malawi, Mauritius, Mozambique, Rwanda, Seychelles, Somalia, South Sudan, Sudan, Tanzania, Uganda, Zambia, and Zimbabwe (Fig 1). A population based crossectional design was conducted. Data for this study were derived from the most recent standard Demographic and Health Surveys (DHS) of the 12 East African countries: Burundi, Comoros, Ethiopia, Kenya, Madagascar, Mozambique, Malawi, Rwanda, Tanzania, Uganda, Zambia, and Zimbabwe. The data spanned from the 2012 Comoros DHS to the 2023 Tanzania DHS. East Africa is of particular interest for this study due to its high reliance on unimproved drinking water sources, coupled with rapid population growth, urbanization, and frequent climate-related challenges such as droughts and floods. The criteria for choosing the most recent surveys included ensuring that the data were up-to-date, representative of the population, and comparable across the selected countries for robust analysis.

A population based crossectional design target a broader and more representative population, often covering entire regions, countries, or large demographic groups. The region encompasses varied landscapes, including arid areas, highlands, and coastal zones, each presenting unique water resource challenges[22].In addition, communal water-sharing norms, governed by customary rules, often dictate how resources are allocated among households, sometimes causing conflicts or inequities[23]. In some cultures, sacred water practices reserve certain sources for religious rituals, restricting their use for essential purposes like drinking[24]. We excluded countries such as Eritrea and Sudan because the outcome variable was unavailable for them. Additionally, countries like Djibouti, Mauritius, Seychelles, Somalia, and South Sudan were excluded due to the lack of DHS data. Therefore, this study included only 12 countries from the total 19 countries. The DHS is a nationally representative survey that provides data for

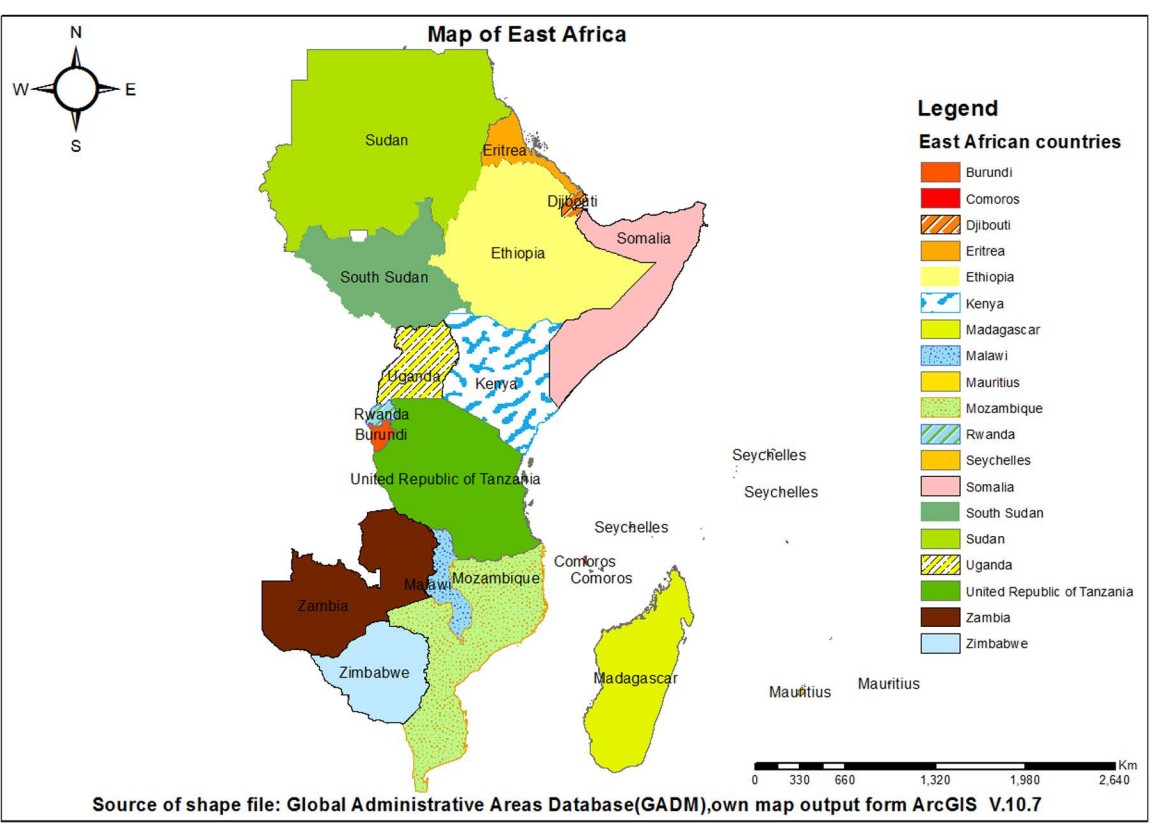

**Fig 1. Map of the study area.**

monitoring indicators of population dynamics, nutrition, and health with large sample size[25]. The permission to use the data was granted from DHS program. DHS data is open source and can be retrieved on the DHS website (https://dhsprogram.com/Data/terms-of-use.cfm).

## Study population and sampling

DHS used a multistage (two stage) cluster sampling technique to recruit the samples using EAs or (Enumeration Areas) or clusters and households as primary and secondary sampling units, respectively. A multistage sampling technique was chosen for the DHS data due to its ability to efficiently capture a representative sample from large, diverse populations across multiple regions. A total of 206,748 households were sampled in 12 East African countries and 8040 clusters was considered for analysis. Burundi, Comoros, Ethiopia, Kenya, Madagascar, Malawi, Mozambique, Rwanda, Tanzania, Uganda, Zambia, and Zimbabwe have 554, 252, 645, 1,692, 657, 850, 619, 500, 629, 697, 545, and 400 clusters, respectively. In DHS there is a probability that some of the regions or countries' were oversampled and some were under-sampled specifically in large countries. Oversampling can exaggerate trends in certain areas, while under sampling may mask disparities or underestimate issues. Therefore, sampling weights were calculated for each household record (HR) of DHS datasets.

## Study variable

**Outcome variable.** According to the WHO/UNICEF joint monitoring programme drinking water sources were categorized as "unimproved" and "improved". Unimproved water sources

include unprotected wells, springs, surface water, vendor-supplied water, and bottled water (if no improved source is available), while improved sources comprise household connections, public standpipes, boreholes, protected wells, springs, and rainwater collection[3,4].

## Data management and analysis

Data management and analysis was performed using Microsoft Excel 2010, STATA 17, ArcGIS software version 10.7, and Kuldorff's SaTScan 10.1 software. Microsoft Excel 2010: Used for initial data cleaning. STATA 17: For statistical weighting and merging data files. ArcGIS software version 10.7: For geospatial analysis and mapping. Kuldorff's SaTScan 10.1: To identify spatial clusters using statistical models.

**Spatial autocorrelation (Global Moran's I) analysis.** Examines the degree to which a set of spatial features and their associated data values tend to be clustered together in space (positive spatial autocorrelation) or dispersed (negative spatial autocorrelation). Positive autocorrelation occurs when Moran's I value is closer to +1, indicating the clustering of similar values. Negative autocorrelation occurs when Moran's I value is closer to -1, indicating a dispersed pattern of dissimilar values. This analysis helps identify patterns, such as whether high values are found near other high values or if they are spread out across the study area[26].This analysis was conducted using ArcGIS software version 10.7.

**Cluster and outlier analysis.** Conducted using the Anselin Local Moran's I statistic, is a spatial analysis method used to identify the presence and location of clusters and outliers in spatial data. This analysis helps understand patterns by indicating areas where high or low values cluster together or where outliers significantly differ from their neighbors[27]. This analysis was employed using ArcGIS software version 10.7.

**Hot spot and cold spot analysis.** Is a spatial analysis technique used to identify areas with statistically significant clusters of high values (hotspots) and low values (cold spots). A commonly used method for this type of analysis is the Getis-Ord Gi* statistic in ArcGIS software version 10.7, known for its robustness in identifying spatial clustering [28]. This analysis was computed using ArcGIS software version 10.7.

**Kriging interpolation analysis.** Kriging is a more advanced geostatistical interpolation technique that incorporates both the distance and the degree of variation between known data points to estimate values at unknown points[29,30]. In this study, we utilized Ordinary Kriging for interpolation analysis. This analysis was conducted using ArcGIS software version 10.7.

**Spatial scan statistical analysis.** We used Kuldorff's SaTScan software v.10.1 for spatial scan statistical analysis with the Bernoulli model in order to determine the purely spatial cluster of unimproved drinking water source in East Africa[31]. The Bernoulli model is a commonly used statistical framework in spatial scan statistics, particularly in spatial cluster detection[32]. It is applied when the data of interest is binary or dichotomous (e.g., cases versus non-cases, presence versus absence)[33]. The Bernoulli model evaluates whether the observed distribution of events is consistent with a null hypothesis of spatial randomness or if significant clusters exist[34]. This model is particularly involves case file, control file and geographic coordinate file. Case file a file includes cluster number and the number of unimproved drinking water source. Control file includes same cluster number with case file and the number of improved drinking water source. Geographic coordinate file includes same cluster number with case file and the geographic coordinates(x, y coordinates). A spatial cluster size threshold of 50% of the total population was applied as the upper limit. Primary and secondary clusters were identified and ranked based on their log-likelihood ratio, with corresponding p-values assigned. Therefore, it will show parts of the East Africa region with high unimproved source of drinking water using high log-likelihood ratio (LLR) and p-value

less than 0.05 to associate with clusters outside of the window. The Log-Likelihood Ratio (LLR) is a statistical measure used to assess the strength of evidence for the presence of spatial clustering or patterns in the data.

### Ethics approval and consent to participate

This study is a secondary analysis of previously collected and published household survey data. As such, it did not require independent ethical approval. However, each included survey obtained ethical clearance from respective national review boards, and informed consent was acquired from all participants. The specific consent procedures varied across surveys. Therefore, ethical approval and informed consent are not applicable to this study.

## Results

### Graduated colour map

The spatial distribution of unimproved drinking water sources across East Africa varied by country. Additionally, the percentage of unimproved water sources per cluster showed significant variation within each country. The percentages of unimproved water sources in various countries are as follows: Burundi (16.81%), Comoros (7.70%), Ethiopia (29.02%), Kenya (23.15%), Madagascar (52.07%), Malawi (13.00%), Mozambique (31.13%), Rwanda (19.51%), Tanzania (24.08%), Uganda (21.90%), Zambia (31.15%), and Zimbabwe (17.73%). As illustrated in (Fig 2 and Fig 3), some areas, such as Madagascar, Zambia, and the coastal regions of Mozambique, showed a high percentage of unimproved water sources.

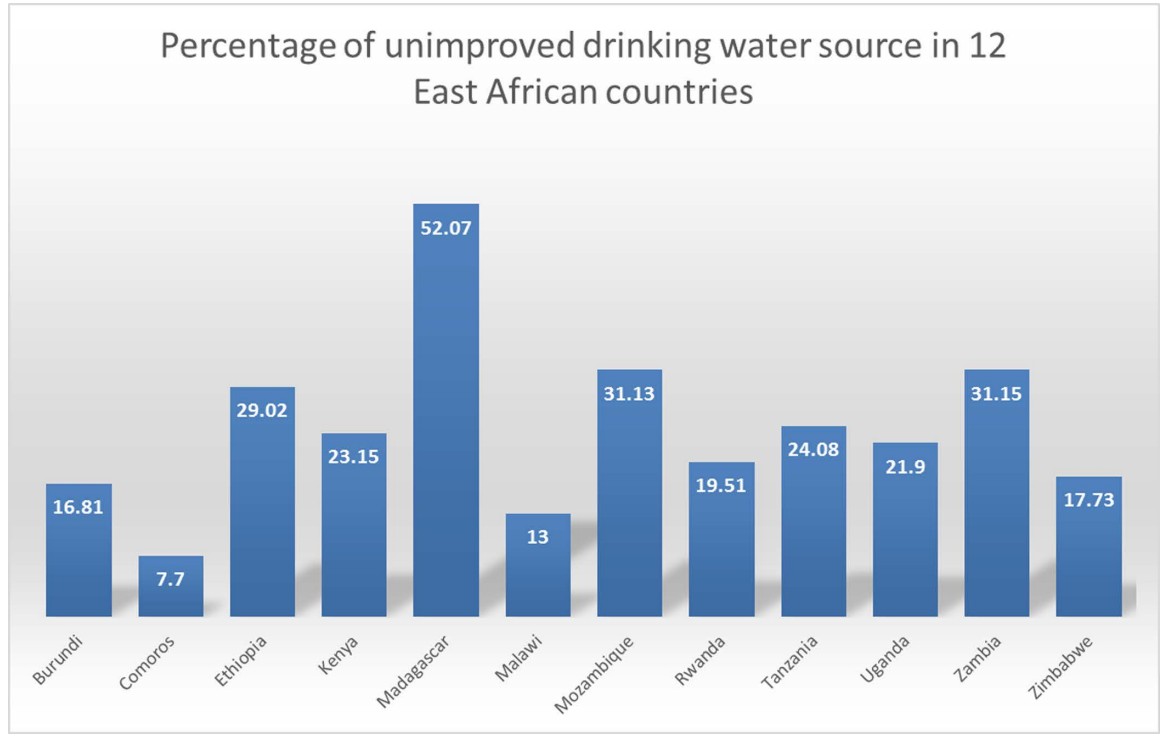

**Fig 2. Percentage of unimproved drinking water sources across East Africa.** DHS data is not available for Sudan, South Sudan, Eritrea, Somalia, Djibouti, Mauritius, and Seychelles.

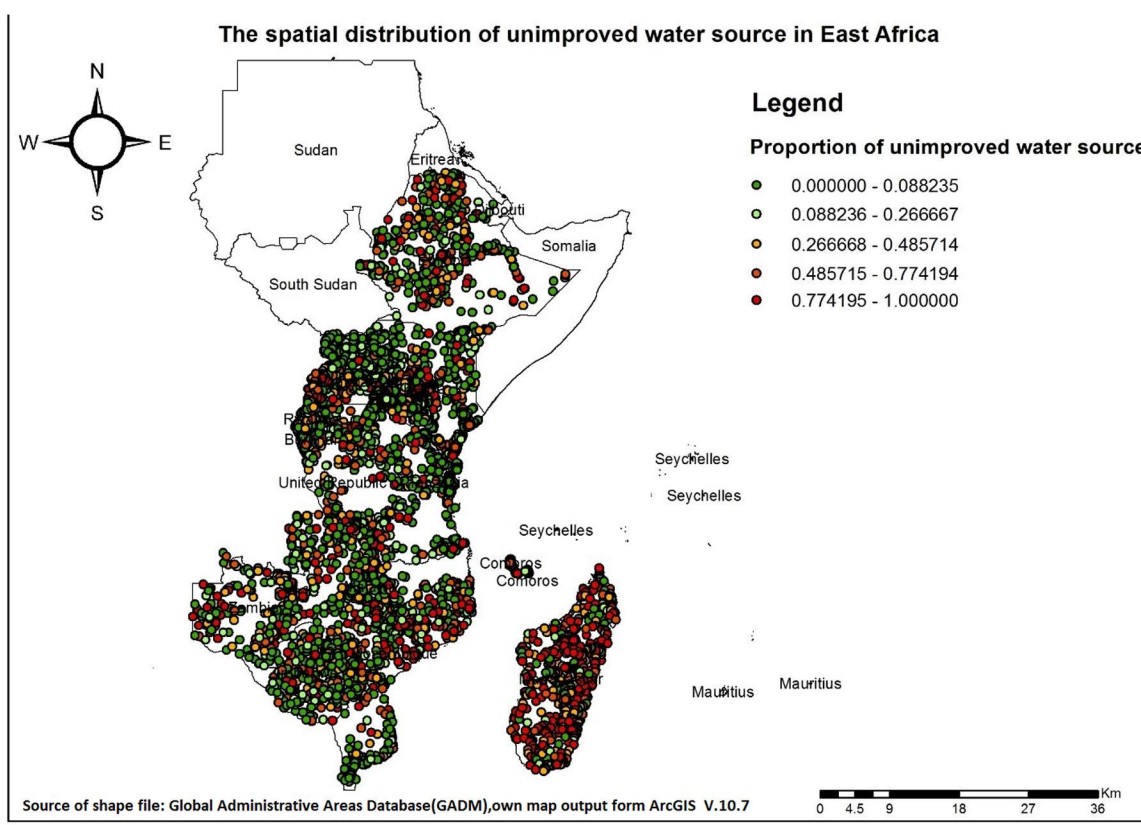

**Fig 3. Spatial distribution of unimproved drinking water sources across East Africa.** DHS data is not available for Sudan, South Sudan, Eritrea, Somalia, Djibouti, Mauritius, and Seychelles.

### Spatial autocorrelation

Spatial analysis using Global Moran's I revealed a significant clustering of unimproved drinking water sources in East Africa (z-score = 87.10087, p < 0.0001). This indicates an extremely low probability (less than 1%) that the observed clustering pattern occurred by chance. The resulting map, with its distinct red and blue clusters, visually confirms this significant spatial pattern (Fig 4).

### Cluster and outlier

The map below illustrates areas of extreme concentration (high-high and low-low) of unimproved drinking water sources in East Africa. Red colored regions, classified as high-high clusters, indicate a high prevalence of unimproved water sources, particularly in Madagascar and parts of Mozambique. Conversely, blue-colored areas represent low-low clusters, where unimproved drinking water sources are less common. These low-low clusters encompass portions of Ethiopia, Kenya, Uganda, Tanzania, Rwanda, Burundi, Malawi, Zimbabwe, Mozambique, and Zambia (Fig 5).

### Hot spot and cold spot

Hot spot and cold spot analysis identified Madagascar, Kenya, and Mozambique as regions with high proportions of unimproved drinking water sources. Conversely, Burundi, Comoros, Ethiopia, Malawi, Rwanda, Tanzania, Uganda, Zambia, and Zimbabwe exhibited lower proportions of unimproved water sources within the region (Fig 6).

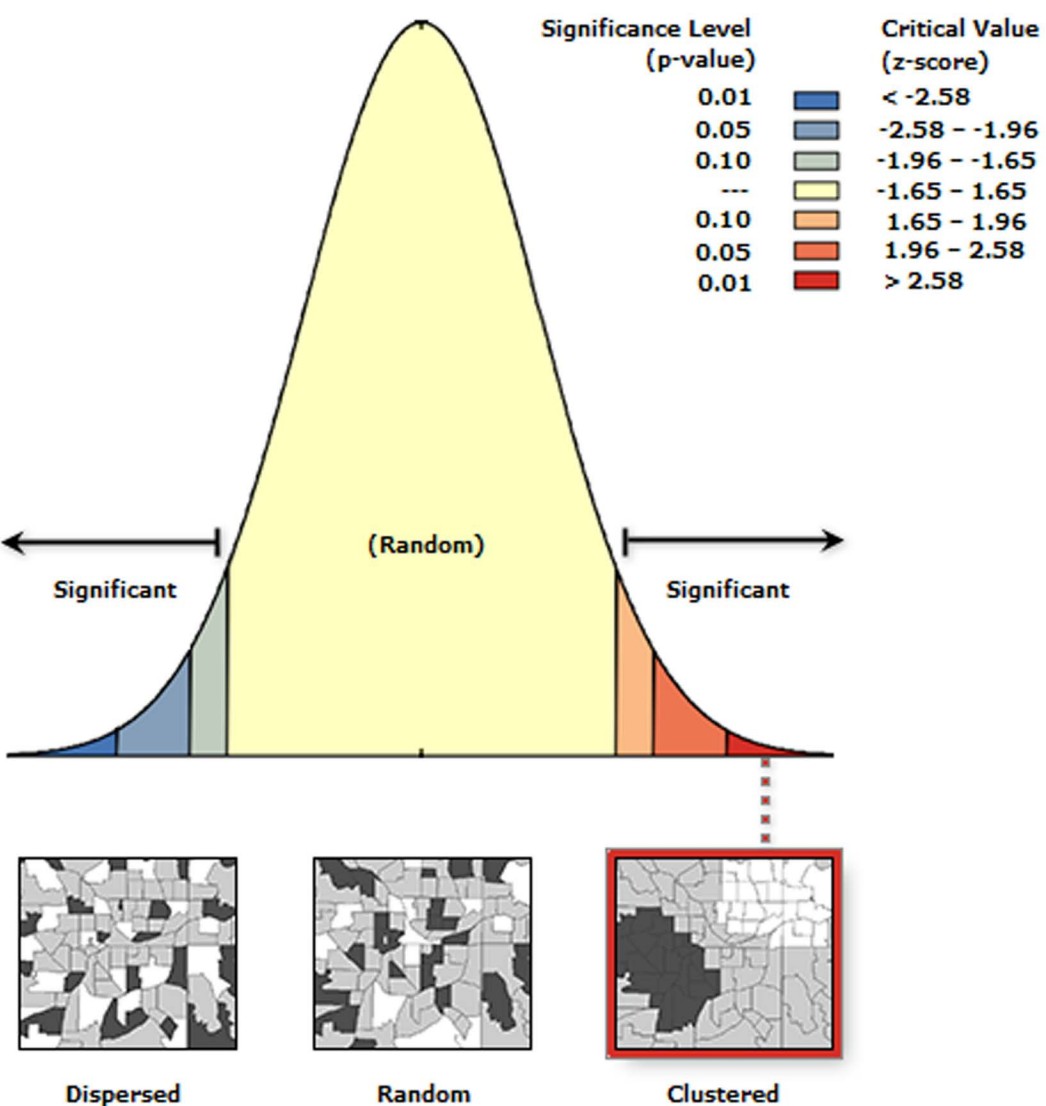

**Fig 4. Spatial autocorrelation analysis of unimproved drinking water sources across East Africa.**

### Kriging interpolation

Spatial Kriging interpolation analysis revealed that areas with higher proportions of unimproved drinking water sources are represented by red on the map, encompassing some parts of Madagascar and Mozambique. Conversely, green colored regions, including Burundi, Comoros, Ethiopia, Kenya, Malawi, Rwanda, Tanzania, Uganda, Zambia, and Zimbabwe, indicate lower levels of unimproved drinking water sources. Kriging has some limitations, such as sparse data, which can sometimes make predictions less reliable (Fig 7).

### Spatial scan statistical

SaTScan spatial analysis of unimproved drinking water sources in East Africa identified one spatial window containing a primary cluster and 166 spatial windows containing secondary

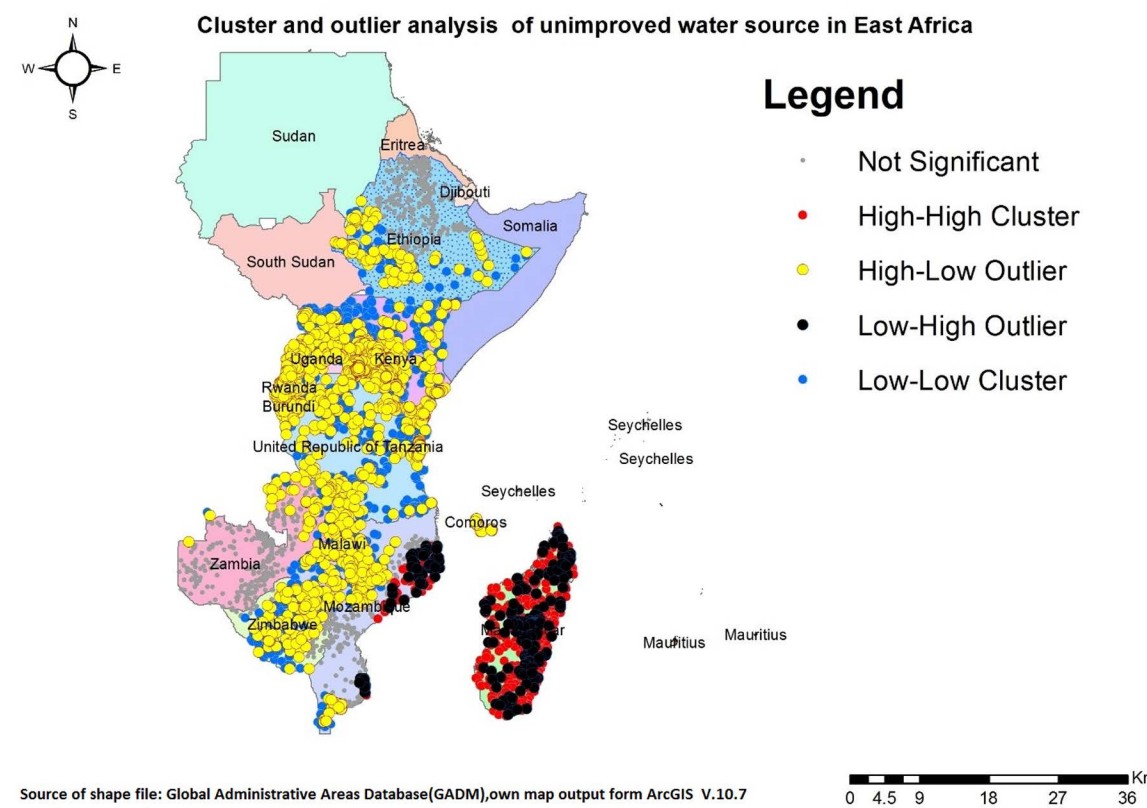

**Fig 5. Cluster and outlier analysis of unimproved drinking water sources across East Africa.**

clusters. The primary clusters covers nearly all parts of Madagascar and certain regions of Mozambique. These clusters exhibited a Log Likelihood Ratio (LLR) of 5221.25 with a p-value less than 0.001. The primary clusters in Madagascar have the highest concentration of unimproved drinking water sources and are typically characterized by severe challenges in accessing safe water. All primary clusters require urgent, large-scale interventions. The secondary clusters were found in all 12 countries (Burundi, Comoros, Ethiopia, Kenya, Madagascar, Malawi, Mozambique, Rwanda, Tanzania, Uganda, Zambia, and Zimbabwe) included in the analysis with LLR spanned from 11.93 to 3135.38. Secondary clusters, on the other hand, are typically smaller and more dispersed. These areas may already have partial access to improved water sources, making their needs less critical than primary clusters (Fig 8).

## Discussion

This study aimed to investigate the geographical variation of unimproved drinking water sources in East African countries using DHS data. The study indicates a significant spatial variation in the distribution of unimproved drinking water sources across East African countries, as evidenced by cluster and outlier analysis, hot and cold spot identification, Kriging, and SaTScan analysis. A similar study also highlights geographic variations in the use of unimproved drinking water sources in Sub-Saharan Africa, emphasizing the need for targeted policies and metrics to address the needs of the most marginalized populations[12]. Higher LLR values in SaTScan analysis indicate stronger clustering of unimproved water sources,

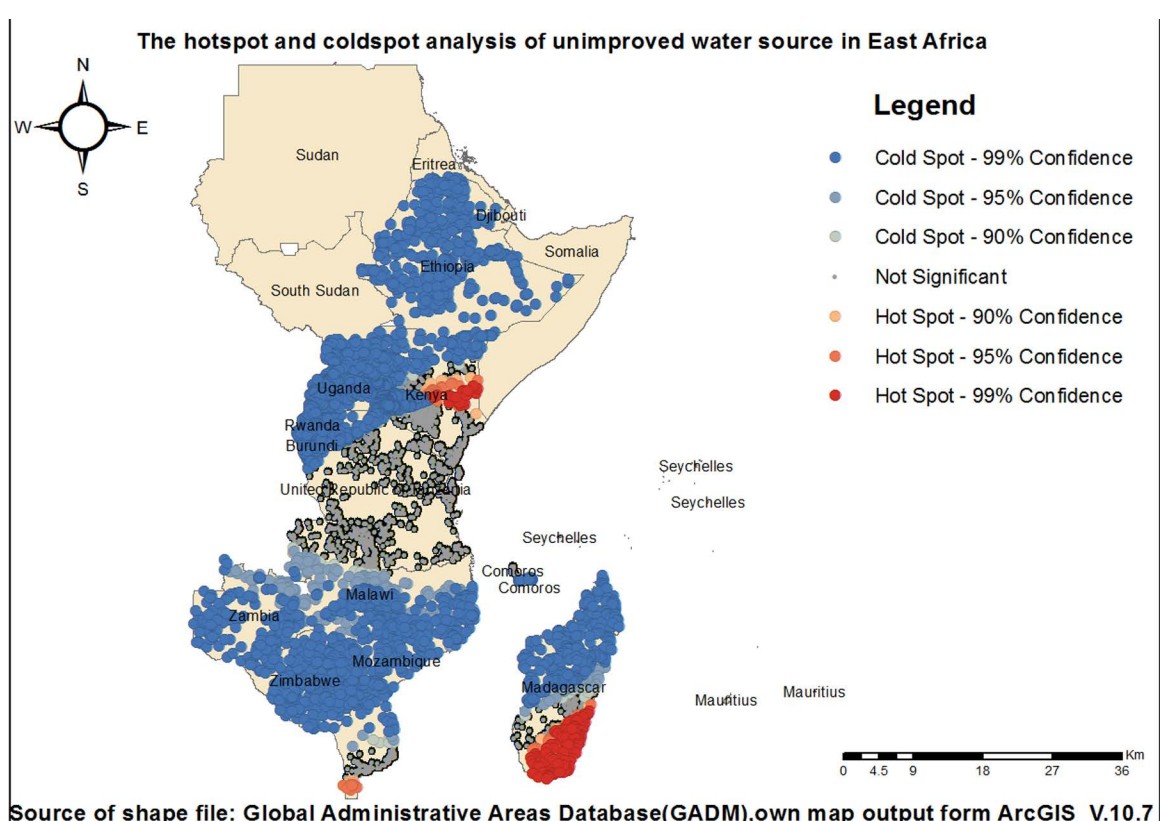

**Fig 6. Hotspot and cold spot analysis of unimproved drinking water sources across East Africa.**

suggesting that a particular area faces significant clean water access challenges. The SaTS-can analysis identified a primary cluster encompassing nearly all of Madagascar and parts of Mozambique, with a Log Likelihood Ratio (LLR) of 5221.25.

This extensive spatial clustering may be attributed to Madagascar's severe water crisis, exacerbated by its extreme vulnerability to climate change[35]. The country frequently suffers from the impacts of climate variability, including recent cyclones, which contribute to water scarcity [36,37]. Moreover, Madagascar is highly vulnerable to projected intensifications in flood risks, rainfall, and drought associated with climate change[38]. A significant portion of Madagascar's population lives in poverty[39], limiting their ability to invest in water infra-structure or alternative water sources. This is compounded by the fact that Madagascar's economy heavily relies on agriculture, a sector highly vulnerable to water scarcity and climate change[38].The interplay of temporal and spatial rainfall disparities further exacerbates water shortages in the region[40].

Coastal areas of Mozambique were also included within the primary cluster. This might be attributed to upstream over-abstraction, pollution, and the intensified impacts of climate change, such as more severe coastal cyclones, increased flooding, and a higher frequency of drought[41,42]. Similar to Madagascar, a substantial portion of Mozambique population lives below the poverty line, limiting their ability to invest in water infrastructure or access improved water services[43].

The SaTScan analysis identified an additional 166 spatial windows containing secondary clusters distributed across all 12 East African countries: Burundi, Comoros, Ethiopia, Kenya, Madagascar, Malawi, Mozambique, Rwanda, Tanzania, Uganda, Zambia, and Zimbabwe.

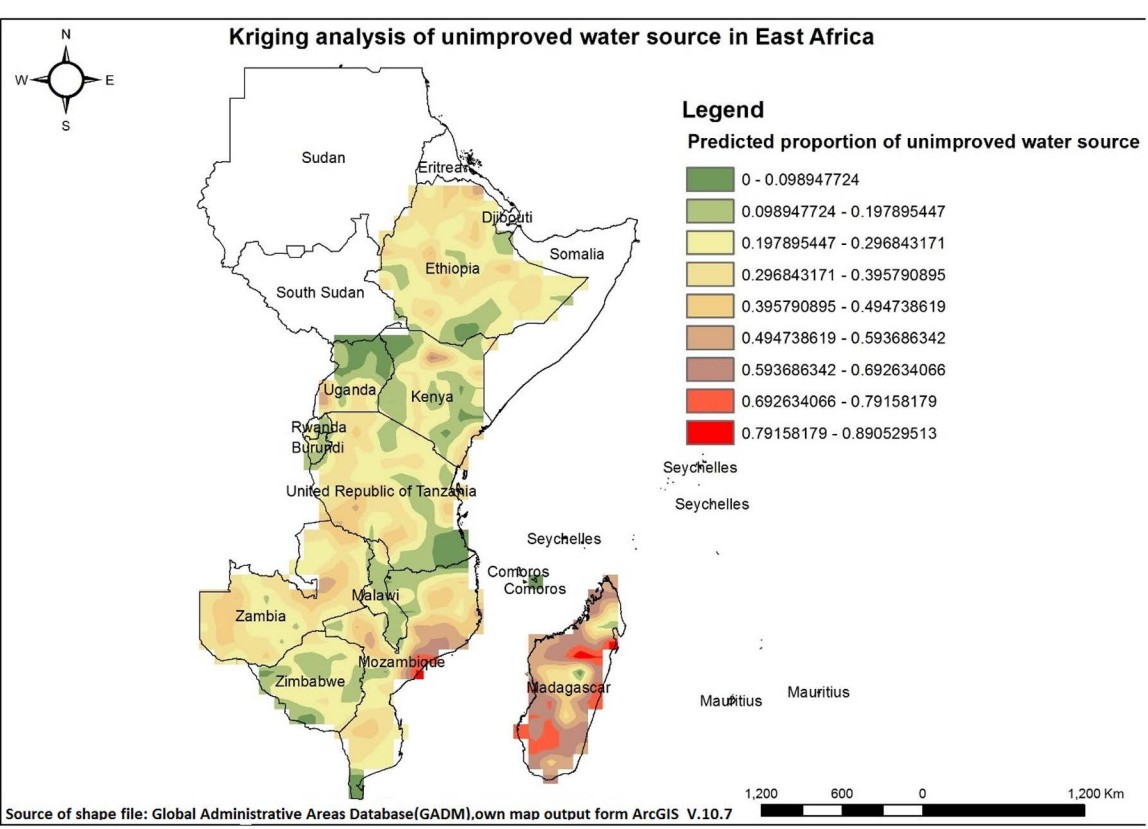

**Fig 7. Kriging analysis of unimproved drinking water sources across East Africa.**

These clusters exhibited Log Likelihood Ratios (LLRs) ranging from 11.93 to 3135.38 having higher LLR in Zambia, Kenya, Tanzania, and Ethiopia. Droughts, seasonal water scarcity, or lack of reliable groundwater sources may exacerbate the reliance on unimproved water in certain regions, contributing to the clustering detected by high LLR values[44,45]. The region's arid and semi-arid climate[46] results in water scarcity and irregular rainfall patterns[47]. In addition, uneven distribution of water resources, coupled with frequent droughts, floods, and cyclones that disrupt infrastructure and contaminate water sources[48]. High poverty rates hinder investments in essential water infrastructure and sanitation, while a significant portion of the population, especially in rural areas, lacks access to basic services including water. Conflict and instability further exacerbate the situation by disrupting water supply systems and impeding development efforts[49]. Inadequate water infrastructure, poor sanitation practices, and inefficient water management compounded by corruption and a lack of political will contribute to the problem[50,51]. Furthermore, over-extraction of water, deforestation, and soil erosion contribute to water scarcity and pollution, further compromising the quality of drinking water sources in the region[52]. Some countries have made significant progress in improving access to clean water. For instance, Rwanda has achieved notable improvements through infrastructure investments, community water systems, national policies recognizing water as a basic right, and partnerships with international organizations [53–55]. Similarly, Uganda has enhanced water access by implementing decentralized management systems, promoting rainwater harvesting, and rehabilitating boreholes to strengthen rural resilience [54,56,57]. Tanzania has also expanded and sustained access to improved water sources through rural

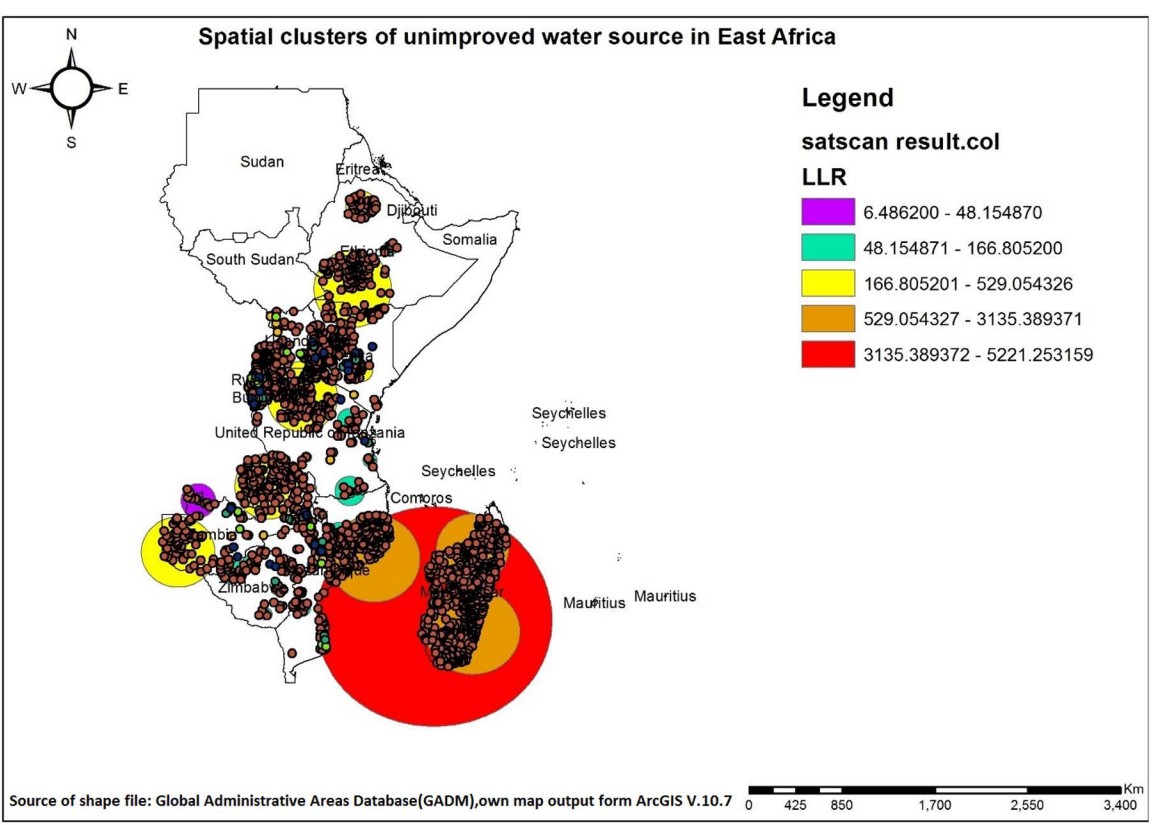

**Fig 8. Spatial scan statistical analysis of unimproved drinking water sources across East Africa.**

water supply initiatives, such as the Water Sector Development Program (WSDP), and active community involvement in managing water points[58,59].

## Strength of the study

The use of advanced spatial analysis techniques enhances the identification of clusters of unimproved water sources, pinpointing areas with the greatest need for intervention. Additionally, the findings offer valuable policy-relevant insights, aiding targeted interventions and supporting progress toward Sustainable Development Goal 6 (Clean Water and Sanitation).

## Limitation of the study

Demographic health survey are typically cross-sectional, providing a snapshot of drinking water status at a specific point in time. Some countries lacked DHS, while others lacked outcome variable this will limit the generalizability across the total 19 East African countries. In addition, there might be geographic gaps in DHS coverage, particularly in remote or conflict-affected areas, limiting the ability to assess drinking water status in these regions. Excluding remote or conflict-affected areas in DHS data can underrepresent vulnerable populations, bias spatial patterns, and miss critical hotspots of unimproved water access. This limits the study's applicability to crisis-prone regions, skews policy implications, and reinforces geographic inequities, reducing its relevance for comprehensive decision-making at regional or national levels.

## Conclusion

This study identified critical clusters, hotspots, and high-high outliers of unimproved drinking water sources across the region, underscoring the need for region-specific interventions. Despite notable progress in achieving water-related targets under the Millennium Development Goals (MDGs) and Sustainable Development Goals (SDGs) in some East African countries[60–62], significant challenges persist in ensuring universal access to improved drinking water sources. Kenya, Mozambique, and Madagascar have been identified as significant hotspot areas. These regions should be prioritized based on the scale and intensity of the problem. Within these hotspots, areas with the largest populations relying on unimproved water sources or those experiencing the most severe health and socioeconomic consequences should be targeted first.

Clusters in Ethiopia, Zambia, Tanzania, Mozambique, Kenya, and Madagascar demand immediate attention to expand water treatment facilities, enhance water distribution networks, and protect water sources from contamination. Building climate resilience through initiatives like rainwater harvesting and improved water storage systems is vital for sustaining water security, particularly in drought-prone areas. Strengthening regulatory frameworks, fostering public-private partnerships, and engaging stakeholders are essential for sustainable progress in East Africa. Governments should lead policy-making, supported by technical assistance and capacity building from organizations like WHO and USAID. NGOs and development banks can contribute advocacy, funding, and expertise, while local communities must be actively involved to ensure lasting outcomes.

Limited financial resources can be addressed through partnerships with global funding agencies, aligning support with local needs. Continuous research, adaptive policies, and robust monitoring mechanisms will ensure accountability and effectiveness in addressing water-related challenges. By aligning efforts and focusing on region-specific needs, East African countries can make significant strides toward achieving universal access to safe and improved drinking water sources. Finally, we recommend further studies to investigate the prevalence and distribution of chronic health conditions associated with prolonged exposure to unsafe water, such as gastrointestinal diseases, stunted growth in children, and long-term cognitive impairments. Integrating health outcome data with spatial analysis could help identify high-risk regions where the impacts of unimproved water sources are most severe. This approach would support the development of targeted public health strategies and interventions, ensuring resources are prioritized for areas in greatest need.

## Author contributions

**Conceptualization:** Lidetu Demoze.

**Data curation:** Lidetu Demoze, Eyob Akalewold, Gelila Yitageasu.

**Formal analysis:** Lidetu Demoze, Kassaw Chekole Adane, Eyob Akalewold, Tenagne Enawugaw, Amensisa Hailu Tesfaye, Gelila Yitageasu.

**Funding acquisition:** Jember Azanaw, Tenagne Enawugaw.

**Investigation:** Lidetu Demoze, Mitkie Tigabie, Gelila Yitageasu.

**Methodology:** Lidetu Demoze, Mitkie Tigabie, Amensisa Hailu Tesfaye.

**Project administration:** Lidetu Demoze.

**Resources:** Lidetu Demoze, Mitkie Tigabie, Gelila Yitageasu.

**Software:** Lidetu Demoze, Kassaw Chekole Adane, Jember Azanaw, Eyob Akalewold, Amensisa Hailu Tesfaye, Gelila Yitageasu.

**Supervision:** Lidetu Demoze, Kassaw Chekole Adane, Jember Azanaw, Tenagne Enawugaw, Amensisa Hailu Tesfaye.

**Validation:** Lidetu Demoze, Jember Azanaw, Eyob Akalewold, Tenagne Enawugaw, Mitkie Tigabie.

**Visualization:** Lidetu Demoze, Eyob Akalewold, Amensisa Hailu Tesfaye.

**Writing – original draft:** Lidetu Demoze.

**Writing – review & editing:** Lidetu Demoze, Jember Azanaw, Amensisa Hailu Tesfaye, Gelila Yitageasu.

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
