## [Decision Letter · Decision Letter 0]

12 Nov 2024

PONE-D-24-33669Spatial analysis of unimproved drinking water source in East Africa: Based on demographic and health survey data, 2012-2022PLOS ONE

Dear Dr. Demoze,

Thank you for submitting your manuscript to PLOS ONE. After careful consideration, we feel that it has merit but does not fully meet PLOS ONE’s publication criteria as it currently stands. Therefore, we invite you to submit a revised version of the manuscript that addresses the points raised during the review process.

Please submit your revised manuscript by Dec 27 2024 11:59PM. If you will need more time than this to complete your revisions, please reply to this message or contact the journal office at plosone@plos.org . Please include the following items when submitting your revised manuscript:

We look forward to receiving your revised manuscript.

Kind regards,

Denekew Bitew Belay, Ph.D

Academic Editor

PLOS ONE

Evaluation of World Bank/French Development Agency financed Urban Water Reform Programme in Lagos Water Corporation (2005-2017) - https://researchportal.bath.ac.uk/en/studentTheses/evaluation-of-world-bankfrench-development-agency-financed-urban-

(Among others)

In your revision ensure you cite all your sources (including your own works), and quote or rephrase any duplicated text outside the methods section. Further consideration is dependent on these concerns being addressed.

3. We note that Figures 1,2,4,5,6 and 7 in your submission contain [map/satellite] images which may be copyrighted. All PLOS content is published under the Creative Commons Attribution License (CC BY 4.0), which means that the manuscript, images, and Supporting Information files will be freely available online, and any third party is permitted to access, download, copy, distribute, and use these materials in any way, even commercially, with proper attribution. For these reasons, we cannot publish previously copyrighted maps or satellite images created using proprietary data, such as Google software (Google Maps, Street View, and Earth). For more information, see our copyright guidelines: http://journals.plos.org/plosone/s/licenses-and-copyright.

a. You may seek permission from the original copyright holder of Figures 1,2,4,5,6 and 7 to publish the content specifically under the CC BY 4.0 license. 

Additional Editor Comments:

The author should to critically follow and address the reviewer comments, which are necessary to improve the current manuscript  to be considered for publication.

Reviewers' comments:

Reviewer's Responses to Questions

**Comments to the Author**

1. Is the manuscript technically sound, and do the data support the conclusions?

Reviewer #1: Yes

Reviewer #2: Partly

Reviewer #3: Partly

Reviewer #4: Yes

Reviewer #5: Partly

Reviewer #6: Yes

2. Has the statistical analysis been performed appropriately and rigorously? 

Reviewer #1: Yes

Reviewer #2: Yes

Reviewer #3: No

Reviewer #4: No

Reviewer #5: No

Reviewer #6: Yes

3. Have the authors made all data underlying the findings in their manuscript fully available?

Reviewer #1: Yes

Reviewer #2: Yes

Reviewer #3: Yes

Reviewer #4: Yes

Reviewer #5: No

Reviewer #6: Yes

4. Is the manuscript presented in an intelligible fashion and written in standard English?

Reviewer #1: Yes

Reviewer #2: No

Reviewer #3: No

Reviewer #4: Yes

Reviewer #5: Yes

Reviewer #6: Yes

5. Review Comments to the Author

Reviewer #1: I do have some questions

1. Why you used spatial cluster size threshold of 50%

2. In Exclusion criteria You said “We excluded some countries because we could not find the outcome variable for them, and some countries did not have DHS.” What does it mean some countries did not have DHS, does it mean Don’t have DHS data set at all or does not have coordinate spatial file? How you manage zero coordinate shape file?

3. Which DHS data set type do you use?

4. .How you measure your outcome variable have you got all categories from WHO/JOINT monitoring program

5. In the recommendation section, you said, “We recommend that East African governments." What are East African governments? Does it mean the distribution is the same for all east African countries? What was the importance of doing hotspot analysis?

Reviewer #2: The article "Spatial analysis of unimproved drinking water sources in East Africa: Based on demographic and health survey data, 2012-2022" is informative but could be refined for clarity, appeal, and precision. Here are some comments and suggestions:

Title:

The phrase “Based on demographic and health survey data” is a bit wordy and could be simplified for conciseness. For example:

“Using Demographic and Health Survey (DHS) data from 2012-2022.”

Abstract:

Conciseness and Redundancy:

Some sentences are too wordy and could be streamlined for clarity. For example:

Instead of “A secondary data analysis was conducted using the most recent demographic and health survey data from 12 East African countries,”

Consider: “We analyzed recent Demographic and Health Survey (DHS) data from 12 East African countries.”

Avoid redundancy. Both “Global Moran’s I” and “spatial scan tests using the Bernoulli model” are mentioned multiple times. Mentioning them once in the methods would suffice.

Highlight Key Findings:

Rather than listing all results, focus on the most significant insights. Mentioning all 167 spatial windows might overwhelm the reader. A clearer takeaway would be:

“Primary clusters were found in Madagascar and coastal Mozambique, with secondary clusters distributed across all 12 countries analyzed.”

Avoid overloading with stakeholders:

The long list of stakeholders (UN, USAID, development banks, NGOs, etc.) could be condensed to improve readability:

“We urge collaboration among governments, international organizations, and NGOs to enhance water infrastructure and ensure sustainable access to clean water.”

Introduction:

Sentence Structure and Grammar:

Avoid run-on sentences to improve clarity. For instance:

“According to the WHO and UNICEF Joint Monitoring Programme, improved water sources are defined as those likely protected from external contamination, particularly fecal matter.” can be rephrased for conciseness:

“Improved water sources are defined by WHO/UNICEF as those protected from external contamination, especially fecal matter.”

Clarify Regional Focus and Data Sources:

The transition from global data to East Africa is logical, but a smoother shift will improve flow. Example:

“While water access has improved globally, sub-Saharan Africa still faces significant challenges, with East Africa reporting 68.7% of its water from unimproved sources.”

Research Objective and Rationale:

The objective could be more concisely stated:

“This study aims to map the spatial distribution of unimproved drinking water sources across 12 East African countries using DHS data from 2012-2022.”

Limitations of Prior Studies:

The mention of the 2024 study on pooled prevalence is valuable, but it should clarify more succinctly how this study differs:

“Previous studies focused on pooled prevalence and associated factors but lacked insights into spatial variation, which this study aims to address.”

Method:

Study Design and Region:

Including the exclusion criteria for some countries strengthens clarity. However, it may be helpful to explicitly mention which excluded countries lacked DHS data or outcome variables.

Study Population and Sampling:

The total sample size (206,748 households) is reported clearly, which helps convey the study's scale. However, providing more detail on how the clusters (8,040) are distributed across the 12 countries would enhance transparency.

Data Management and Software:

The use of various software tools (Excel, STATA, ArcGIS, and SaTScan) is comprehensive. A small improvement would be to describe briefly the purpose of each tool. For example:

Microsoft Excel: Used for initial data cleaning.

STATA: For statistical weighting and merging data files.

ArcGIS: For geospatial analysis and mapping.

SaTScan: To identify spatial clusters using statistical models.

Including version numbers of the software is good practice. Make sure the spellings are consistent (e.g., “Kuldorff’s SaTScan”).

Spatial Analysis Methods:

Moran’s I and Anselin Local Moran’s I: These are well explained. Clarifying the interpretation of positive/negative autocorrelation for a non-technical audience might improve accessibility.

Hotspot/Coldspot Analysis: The Getis-Ord Gi* method is mentioned appropriately, but it may help to specify why this method was chosen over others (e.g., due to its robustness in spatial clustering).

Kriging: The description is concise, but it may be helpful to mention if ordinary or universal Kriging was used, as they differ slightly in assumptions and applications.

Results:

Graduated Color Map:

Potential Improvement: Add quantitative summary statistics (e.g., the median or range of proportions per country) to supplement the map. Also, briefly describe any patterns by geography, like whether coastal areas tend to have better access than inland regions.

There is a country or a region that has no data; please put those regions that have no data on the legend for figure two.

Kriging Interpolation Analysis:

Potential Improvement: Mention limitations of Kriging, such as areas where data might be sparse and the predictions less reliable. Also, consider discussing areas where predictions diverge from expected trends, prompting further investigation.

Spatial Scan Statistical Analysis (SaTScan):

Potential Improvement: Explain the significance of primary and secondary clusters. For example, what distinguishes the primary cluster in Madagascar from the other regions? Also, briefly discuss why certain secondary clusters may require targeted but less intensive interventions.

Discussion:

Linking Results to Causes:

Potential Improvement: Strengthen the connection between your statistical results and the identified causes. For instance, mention how the LLRs reflect the severity of water issues or highlight how coastal vs. inland disparities impact the results.

Citing data sources and evidence:

Potential Improvement: Expressly relate your discussion points to the spatial analysis findings. For example, link specific clusters to rainfall patterns or flood-prone areas, referring to maps or figures.

Secondary Clusters: Further Exploration:

Potential Improvement: Provide more detailed insights into variations across secondary clusters. Are there clusters that are unexpectedly severe or others that show gradual improvement? A more granular discussion can deepen the reader’s understanding.

Balanced Discussion:

Potential Improvement: Include positive developments, if applicable. For example, mention countries or regions showing progress or resilience in managing unimproved water sources. This adds nuance and balance to the discussion.

Conclusion

The conclusion effectively summarizes the findings and offers useful recommendations. However, it can be improved by:

Reducing redundancy,

Providing more region-specific recommendations,

Balancing the tone with some acknowledgment of progress,

Offering clearer roles for stakeholders, and

Emphasizing the importance of ongoing monitoring and adaptation.

These improvements will make the conclusion more impactful, actionable, and aligned with the study’s insights.

What is the strength of your article? Please mention.

Reviewer #3: this paper add no value to scientific community doing spatial analysis and knowing only the sites of unimproved drinking water source will not provide solutions and another analysis that is multilevel mixed effect model is needed so at this adage I recommend to add multilevel mixed effect model and write and submit again to the journal so that what factors lead to unimproved drinking water source occurrence is very important

Reviewer #4: Dear editor, thank you for the opportunity to invite “Spatial analysis of unimproved drinking water source in East Africa: Based on demographic and health survey data”. The title is very interesting across east Africa. However, the major limitations of these study is without any statistical model to demonstrate any methodological perspectives for unimproved drinking water source. There is no nobility of study. But, only show the spatial patterns in terms of descriptive statistics. So, these study is better to do again in terms of any statistical model considerations. And also, the authors better to consider machine learning approach in addition to any model. Then, this paper requires major modifications before publication.

My point of view, the following point is issued.

1. In abstract part:

Background: background is too large. So, the authors better to minimize the statements again.

Methods: well. But, better to add inferential statistics or any statistical model in terms of machine learning approach.

Results: good. But, this study is better to consider with statistical model to determine different determinates across east Africa in addition to this work.

Conclusions: Conclusions is well organized and a good part of messages related with these results. However, these conclusions part is stronger in terms of statistical model approach in addition to these descriptive nature of the study.

2. In Introduction part

I appreciated the structure of each paragraph in terms of global to regional. But, there is some grammatical, punctuation, capitalizations problems.

The gap of study is not enough. There is no novel of study. Then, the authors more clarify consider the gap/ nobility of these study.

3. Method

Better to say Material and Methods than “Method”. Study area, design and data source and Study population and sampling is well organized especially in figure.

The authors had outcome variable. But, there was no any statistical model. So, why do not use it???

Generally, the point in methods section was well.

The authors must be check Ethics approval and consent to participate according to journal guidelines. Because these is under declaration sections.

So, the overall expression under the method section is descriptive statistics. These study is more precise and attractive to consider inferential statistics or any statistical model.

4. Results

The descriptive part in result section is well. But, the authors lost any statistical model under inferential statistics.

5. Discussion: The discussion part is more/less better. But, the authors better to compare and contrast these result with previous studies.

6. Conclusion: the conclusion part is enough based on authors methodological approaches. But, more strong in terms of statistical model.

7. Reference: The reference citation style must be check again according to the journal guidelines.

General comments

 These study presents the results of original research across east Africa. But, these study was done in different east Africa countries separately, especially Ethiopia by using different statistical model.

 Results reported have not been published elsewhere.

 The descriptive analyses are performed to a high technical standard and are described in sufficient detail.

 Conclusions are presented in an appropriate fashion and are supported by the data.

 The article is presented in an intelligible fashion and is written in Standard English. But, there is some grammatical, punctuation and capitalization issues in inside these paper.

 The research meets all applicable standards for the ethics of experimentation and research integrity.

 The article adheres to appropriate reporting guidelines and community standards for data availability.

 In my point of view, this paper is near to publication after major modifications as some point mentioned in above.

Reviewer #5: Review Comments to the Author

The manuscript titled "Spatial Analysis of Unimproved Drinking Water Sources in East Africa: Based on Demographic and Health Survey Data, 2012-2022" addresses a critical public health issue. The analysis of unimproved drinking water sources is timely and relevant, given the significant health implications associated with such sources in East Africa. Below are my detailed comments and suggestions for improvement:

1. Technical Soundness:

o Overall, the manuscript presents a technically sound piece of research. The authors employ robust statistical methods, including Global Moran's I and spatial scan statistics, which enhance the validity of the findings. However, some aspects of the methodology could be clarified, particularly regarding the Bernoulli model used in the spatial scan statistical tests. A more detailed explanation would improve the reader's understanding.

2. Data Presentation:

o The results are presented clearly, but the addition of visual elements such as maps and graphs could significantly enhance comprehension. Visual aids will help convey the spatial distribution of unimproved water sources more effectively.

3. Contextual Background:

o While the background section provides a solid foundation, incorporating more context regarding the socio-economic implications of unimproved water sources would deepen the understanding of the research's significance.

4. Limitations:

o The limitations section is somewhat brief. A more comprehensive discussion on how the study's generalizability might be affected by excluding certain remote or conflict-affected areas would provide a more balanced perspective. Addressing potential biases in the data or challenges faced during the analysis is crucial for transparency.

5. Future Research Suggestions:

o I recommend including suggestions for future research, such as exploring the effectiveness of specific interventions or examining the long-term health impacts of unimproved water sources. This could guide subsequent studies in this important area.

6. Ethical Considerations:

o I have no concerns regarding dual publication or research ethics. The research appears to adhere to ethical guidelines, and the authors have adequately addressed the use of data from demographic and health surveys.

7. Conclusion and Recommendations:

o The recommendations provided are actionable and address the critical need for collaboration among stakeholders. However, highlighting successful case studies or ongoing efforts within East Africa that align with these recommendations would offer a practical perspective and underscore the feasibility of proposed actions.

In summary, while the manuscript is well-structured and addresses an important public health challenge, it would benefit from refinements in methodological clarity, contextual background, and data presentation. I believe that addressing these points will enhance the manuscript's impact on the understanding and management of unimproved drinking water sources in East Africa.

Reviewer #6: The study's goal is to figure out the spatial distribution of unimproved drinking water sources in East Africa. This study found a significant spatial clusters, hotspots, and outliers of unimproved drinking water sources in several East African countries. The study identified significant spatial clusters, hotspots, and outliers of unimproved drinking water sources in East Africa, and recommends that these areas be prioritized for expanding water treatment facilities, improving water distribution, and protecting water sources, with collaboration from East African governments, international organizations, and other stakeholders. Spatial clusters of unimproved drinking water sources were detected throughout East Africa, indicating a major global clustering tendency. A primary cluster covering nearly all of Madagascar and parts of Mozambique's coastal regions was identified, exhibiting a high Log Likelihood Ratio (LLR) of 5221.25 and a p-value < 0.000. Secondary clusters were found across all 12 countries included in the analysis, with LLRs ranging from 11.93 to 3135.38 and p-values < 0.05.

The paper is well-structured and addresses an important public health issue with sound analytical methods. Minor revisions to improve clarity, consistency, and the inclusion of additional visuals would elevate the paper further.

Strengths of the article

The paper provides a thorough introduction, effectively contextualizing the importance of access to safe drinking water and the health implications of unimproved sources. This sets the stage for the study's significance. The use of various spatial analysis tools, including ArcGIS and SaTScan, demonstrates a strong and multifaceted approach to analyzing the spatial distribution of unimproved drinking water sources. The presentation of findings with specific statistics.

Areas for improvement

Some portions, particularly the introduction and methodology, might be condensed to improve readability. Breaking down long sentences and reducing difficult phrases would make the text more understandable. Ensure consistency when referring to tools and analysis. For example, utilizing consistent language for software versions and statistical tests promotes clarity. While the methods are thorough, including a simple summary table or flowchart could help readers grasp the analysis workflow, particularly those unfamiliar with spatial statistics. Including brief comparisons or context from similar research in other locations may help to highlight the specific elements of East Africa's difficulties and deepen the conversation. Clarifying which identified hotspots should be prioritized based on severity or population impact could make the recommendations more actionable. The suggestions for future research could be expanded to include potential methods or additional data sources that could refine or complement the current findings.

6. PLOS authors have the option to publish the peer review history of their article (what does this mean? ). If published, this will include your full peer review and any attached files.

**Do you want your identity to be public for this peer review?** For information about this choice, including consent withdrawal, please see our Privacy Policy .

Reviewer #1: **Yes: ** Belayneh Jejaw Abate

Reviewer #2: No

Reviewer #3: No

Reviewer #4: **Yes: ** Nurye Seid Muhie

Reviewer #5: **Yes: ** Sena Adugna Beyene

Reviewer #6: No

---

## [Author Response · Author response to Decision Letter 0]

9 Dec 2024

Dear Editor,

Some of the comments provided by the reviewers, particularly reviewers 3 and 4, are outside the scope of the study. Kindly take this into consideration during the review process.

---

## [Decision Letter · Decision Letter 1]

2 Jan 2025

PONE-D-24-33669R1Spatial analysis of unimproved drinking water source in East Africa: Using Demographic and Health Survey (DHS) data from 2012-2023PLOS ONE

Dear Dr. Demoze,

Thank you for submitting your manuscript to PLOS ONE. After careful consideration, we feel that it has merit but does not fully meet PLOS ONE’s publication criteria as it currently stands. Therefore, we invite you to submit a revised version of the manuscript that addresses the points raised during the review process.

We look forward to receiving your revised manuscript.

Kind regards,

Denekew Bitew Belay, Ph.D

Academic Editor

PLOS ONE

Journal Requirements:

Additional Editor Comments:

There are still a few corrections needed. Please pay attention and make the necessary changes to your manuscript.

Reviewers' comments:

Reviewer's Responses to Questions

**Comments to the Author**

1. If the authors have adequately addressed your comments raised in a previous round of review and you feel that this manuscript is now acceptable for publication, you may indicate that here to bypass the “Comments to the Author” section, enter your conflict of interest statement in the “Confidential to Editor” section, and submit your "Accept" recommendation.

Reviewer #1: All comments have been addressed

Reviewer #2: All comments have been addressed

Reviewer #4: All comments have been addressed

2. Is the manuscript technically sound, and do the data support the conclusions?

Reviewer #1: Yes

Reviewer #2: Yes

Reviewer #4: Yes

3. Has the statistical analysis been performed appropriately and rigorously? 

Reviewer #1: Yes

Reviewer #2: Yes

Reviewer #4: Yes

4. Have the authors made all data underlying the findings in their manuscript fully available?

Reviewer #1: (No Response)

Reviewer #2: Yes

Reviewer #4: Yes

5. Is the manuscript presented in an intelligible fashion and written in standard English?

Reviewer #1: Yes

Reviewer #2: No

Reviewer #4: Yes

6. Review Comments to the Author

Reviewer #1: (No Response)

Reviewer #2: I have carefully assessed the manuscript and provided detailed feedback to address the points requiring clarification and improvement.

Overall, I found the study was well-conducted and addressed my comments. I have seen all areas, where further refinement could enhance the manuscript’s clarity and impact.

Please find my review attached to this email for your reference. I hope my comments contribute to improving the quality of the manuscript. Should you require any additional clarification or further input from me, please feel free to contact me.

Title Spatial analysis of unimproved drinking water source in East Africa: Using Demographic and Health Survey (DHS) data from 2012-2023 (use sources)

Abstract

1. The first letter of each keyword must be a Capital letter.

2. data from 12 East African countries. Please list the countries included in your study.

3. A total of 206,748 households were sampled in 12 East African countries… Did you use sampling weight?

4. Data management and analysis were conducted using Microsoft Excel 2010, STATA 17, ArcGIS software version 10.7, and Kuldorff’s SaTScan 10.1. Excel was used for initial data cleaning, STATA for statistical weighting and data merging, ArcGIS software version 10.7 for geospatial analysis and mapping, and Kuldorff’s SaTScan 10.1 for identifying spatial clusters. There is no need to list the software you have used here; you can list the software you have used for data analysis in the methods and material section. Simply list the methods you have used for data analysis here.

5. Your conclusion is vast, please give your conclusion based on your result.

Reviewer #4: miner revision of the manuscript Spatial analysis of unimproved drinking water source in East Africa: Using Demographic and Health Survey (DHS) data from 2012-2023

7. PLOS authors have the option to publish the peer review history of their article (what does this mean? ). If published, this will include your full peer review and any attached files.

**Do you want your identity to be public for this peer review?** For information about this choice, including consent withdrawal, please see our Privacy Policy .

Reviewer #1: **Yes: ** Belayneh Jejaw Abate

Reviewer #2: No

Reviewer #4: No

---

## [Author Response · Author response to Decision Letter 1]

3 Jan 2025

Responses to the review’s comments

Dear PLOS ONE editorial team,

Thank you for giving us the opportunity to submit a revised draft of the manuscript, and we would also like to thank you for your crucial comments on our paper (Manuscript ID: PONE-D-24-33669 R1). We are very concerned and have combined all the suggested comments provided, which we believe strengthen our paper, and we hope this will render our paper eligible for consideration for publication in your reputed journal. We appreciate the time and effort that you and the reviewers dedicated to providing feedback on our manuscript and are grateful for the insightful comments and valuable improvements to our paper for publication.

The authors would like to inform you that we have addressed the comments and recommendations of the handling editor point by point. In addition, throughout our revision, we made our best corrections too. All changes made to the original version are highlighted using tracking changes and attached as “Revised Manuscript with Track Changes”. The unmarked copy of the manuscript is also attached as “Manuscript”. In addition, please see below a rebuttal letter that responds to each point raised by the handling editor, and this letter is also attached to the submission as “Response to Reviewers”.

Response to editor’s comments

Comments from Reviewer #2:

1. I have carefully assessed the manuscript and provided detailed feedback to address the points requiring clarification and improvement. Overall, I found the study was well-conducted and addressed my comments. I have seen all areas, where further refinement could enhance the manuscript’s clarity and impact. Please find my review attached to this email for your reference. I hope my comments contribute to improving the quality of the manuscript. Should you require any additional clarification or further input from me, please feel free to contact me.

Author’s response: Dear reviewer, thank you very much for your insightful questions, valuable suggestions, and thoughtful recommendations. Your feedback has significantly enhanced the overall quality of the manuscript.

2. The first letter of each keyword must be a Capital letter.

Author’s response: Dear reviewer, thank you very much for your comment. We have made corrections accordingly.

3. Data from 12 East African countries. Please list the countries included in your study.

Author’s response: Dear reviewer, thank you very much for your comment. We have made corrections accordingly.

4. A total of 206,748 households were sampled in 12 East African countries… Did you use sampling weight?

Author’s response: Dear reviewer, thank you very much for your question. Yes, we used sampling weights, and we have already provided a detailed explanation in lines 147 to 151.

5. Data management and analysis were conducted using Microsoft Excel 2010, STATA 17, ArcGIS software version 10.7, and Kuldorff’s SaTScan 10.1. Excel was used for initial data cleaning, STATA for statistical weighting and data merging, ArcGIS software version 10.7 for geospatial analysis and mapping, and Kuldorff’s SaTScan 10.1 for identifying spatial clusters. There is no need to list the software you have used here; you can list the software you have used for data analysis in the methods and material section. Simply list the methods you have used for data analysis here.

Author’s response: Dear reviewer, thank you very much for your comment. We have made corrections accordingly.

6. Your conclusion is vast, please give your conclusion based on your result.

Author’s response: Dear reviewer, thank you very much for your comment. We have made the necessary corrections. The majority of the conclusions section was recommendations, which led to its length. We have minimized it as much as possible while ensuring that the feedback from other reviewers is not compromised.

Comments from Reviewer #4:

1. Miner revision of the manuscript Spatial analysis of unimproved drinking water source in East Africa: Using Demographic and Health Survey (DHS) data from 2012-2023

Author’s response: Dear reviewer, thank you very much for your effort to improve the overall quality of the manuscript.

---

## [Editor Report · Decision Letter 2]

13 Jan 2025

Spatial analysis of unimproved drinking water source in East Africa: Using Demographic and Health Survey (DHS) data from 2012-2023

PONE-D-24-33669R2

Dear Dr. Demoze,

We’re pleased to inform you that your manuscript has been judged scientifically suitable for publication and will be formally accepted for publication once it meets all outstanding technical requirements.

Kind regards,

Denekew Bitew Belay, Ph.D

Academic Editor

PLOS ONE
---

## [Editor Report · Acceptance letter]

PONE-D-24-33669R2

PLOS ONE

Dear Dr. Demoze,

I'm pleased to inform you that your manuscript has been deemed suitable for publication in PLOS ONE. Congratulations! Your manuscript is now being handed over to our production team.

Kind regards,

on behalf of

Dr. Denekew Bitew Belay

Academic Editor

PLOS ONE